# Extracellular Vesicles—A Source of RNA Biomarkers for the Detection of Breast Cancer in Liquid Biopsies

**DOI:** 10.3390/cancers15174329

**Published:** 2023-08-30

**Authors:** Pawel Zayakin, Lilite Sadovska, Kristaps Eglītis, Nadezhda Romanchikova, Ilze Radoviča-Spalviņa, Edgars Endzeliņš, Inta Liepniece-Karele, Jānis Eglītis, Aija Linē

**Affiliations:** 1Latvian Biomedical Research and Study Centre, Ratsupites Str. 1, k-1, LV-1067 Riga, Latvia; pawel@biomed.lu.lv (P.Z.); lilite@biomed.lu.lv (L.S.); nadezda.romancikova@biomed.lu.lv (N.R.); edgars.endzelins@biomed.lu.lv (E.E.); 2Latvian Oncology Center, Riga Eastern Clinical University Hospital, LV-1038 Riga, Latvia; kristaps.eglitis@aslimnica.lv (K.E.);; 3Genera Ltd., Mārupes Str. 22, LV-1002 Riga, Latvia; ilze@genera.lv; 4Department of Pathology, Riga Stradins University, LV-1007 Riga, Latvia

**Keywords:** extracellular vesicles, breast cancer, RNA sequencing, snoRNA, liquid biopsy, diagnostic biomarker, mitochondria-derived EVs

## Abstract

**Simple Summary:**

Extracellular vesicles (EVs) are nano-sized membrane-bound vesicles released into body fluids by various cell types, including cancer cells, that carry molecular cargo derived from their parental cells. Therefore, they are gaining attention as carriers of cancer biomarkers for liquid biopsies. Here, we sought to identify cancer-derived RNA biomarkers in plasma EVs from breast cancer (BC) patients. We reasoned that the amount of cancer-derived RNAs should decrease after surgical removal of the tumor; therefore, we compared the composition of EV RNA in BC patients at the time of diagnosis and 7 days after surgery and in cancer-free females. The results showed that the fractions of miRNAs, snRNAs, snoRNAs, and tRFs were increased, but the fraction of lncRNAs was decreased in BC EVs compared with healthy controls. We identified biomarker candidates among different RNA biotypes and created biomarker models that can detect BC and inform tumor estrogen receptor and HER2 status with remarkably high accuracy.

**Abstract:**

Over the past decade, extracellular vesicles (EVs) have emerged as a promising source of cancer-derived RNAs for liquid biopsies. However, blood contains a pool of heterogeneous EVs released by a variety of cell types, making the identification of cancer RNA biomarkers challenging. Here, we performed deep sequencing of plasma EV RNA cargo in 32 patients with locally advanced breast cancer (BC) at diagnosis and 7 days after breast surgery and in 30 cancer-free healthy controls (HCs). To identify BC-derived RNA biomarkers, we searched for RNAs that had higher levels in BC EVs at the time of diagnosis compared with HCs and decreased after surgery. Data analysis showed that the fractions of miRNAs, snRNAs, snoRNAs, and tRFs were increased, but the fraction of lncRNAs was decreased in BC EVs as compared to HCs. BC-derived biomarker candidates were identified across various RNA biotypes. Considered individually, they had very high specificity but moderate sensitivity for the detection of BC, whereas a biomarker model composed of eight RNAs: SNORD3H, SNORD1C, SNORA74D, miR-224-5p, piR-32949, lnc-IFT-122-2, lnc-C9orf50-4, and lnc-FAM122C-3 was able to distinguish BC from HC EVs with an AUC of 0.902 (95% CI = 0.872–0.931, *p* = 3.4 × 10^−9^) in leave-one-out cross-validation. Furthermore, a number of RNA biomarkers were correlated with the ER and HER2 expression and additional biomarker models were created to predict hormone receptor and HER2 status. Overall, this study demonstrated that the RNA composition of plasma EVs is altered in BC patients and that they contain cancer-derived RNA biomarkers that can be used for BC detection and monitoring using liquid biopsies.

## 1. Introduction

Breast cancer (BC) is the most frequent cancer in women worldwide, with an estimated 2.1 million new cases and over 600,000 deaths annually [1] and its global incidence is steadily rising [2]. If BC is detected at an early stage when the cancer is localized only in the breast, 70–80% of the patients can be cured; however, the remaining patients will develop distant metastases within 10 years [3]. Unfortunately, a significant fraction of cases are diagnosed when cancer has already spread to distant organs and then it is considered to be incurable [2]. BC is a profoundly heterogeneous disease with respect to the gene expression profiles, mutational landscape, histology, metastasis patterns, response to treatment, and patient outcomes [4]. Based on the gene expression profiles, five intrinsic molecular subtypes (luminal A, luminal B, HER2-enriched, basal-like, and Claudin-low) have been established. These subtypes differ in their clinical course and treatment sensitivity; therefore, they provide important prognostic and predictive information for patient stratification [5,6]. However, due to the cost and logistic issues, molecular assays for determining the subtypes are not available for a large portion of patients; therefore, in the current clinical practice, surrogate definitions of the subtypes based on immunohistochemical detection of estrogen receptor (ER), progesterone receptor (PR), HER2, and Ki-67 are often used for making treatment decisions [6].

Liquid biopsies have emerged as an alternative to tissue biopsies. These are samples of blood or other body fluids, where circulating tumor cells or cancer-derived molecules such as cell-free tumor DNA or RNA are tested [7,8]. In BC, liquid biopsies could be exploited for the early detection and diagnosis, prediction or monitoring of response to treatment, and early detection of relapse. Extracellular vesicles (EVs) are increasingly recognized as a source of cancer-derived biomarkers in liquid biopsies [9]. EVs are vesicles that are naturally released from cells and are enclosed by a plasma membrane and cannot replicate [10]. Three main types of EVs have been defined based on the mode of biogenesis: exosomes, microvesicles, and apoptotic bodies. They differ in their size, membrane composition, molecular cargo, and likely have distinct functions [11]. In addition, two types of nonvesicular extracellular particles (NVEPs), exomeres and supremeres, have been recently discovered and shown to carry extracellular RNA [12,13]. Hence, human blood contains a complex mixture of various EV and NVEP types produced by multiple cell types. 

EVs isolated from cancer patients’ blood have been shown to carry oncoproteins, mutated mRNA and DNA fragments, and cancer-associated miRNA signatures [14,15,16,17], which has led to the idea that the analysis of molecular content of EVs could inform about the presence, molecular profile, and behavior of cancer. However, the fraction of cancer-derived EVs is likely to be low and/or variable among patients; hence, cancer-derived molecules may be highly diluted and the molecular signatures may be blurred by normal tissue-derived EVs.

EV-RNA sequencing studies have revealed that EVs contain fragmented mRNAs and a variety of non-coding RNA biotypes [18,19,20]. Several studies have explored the RNA cargo of plasma EVs in BC patients; however, the majority of them were focused on miRNAs [21,22,23,24], mRNAs, and lncRNAs [25,26], whereas the diagnostic significance of other RNA biotypes has not been systematically explored. Moreover, the majority of the studies have been designed as case–control studies. Although the RNA biomarkers identified in these studies are significantly associated with the disease status or prognosis, the tissue origin of these biomarkers remains unknown—RNAs that have higher levels in cancer patients than controls might be produced by cancers themselves or by other cell types responding to the presence of cancer or inflammation, whereas the origin of RNAs that have higher levels in cancer-free controls than patients is entirely obscure.

Therefore, in the current study, we took a different approach. We reasoned that the levels of cancer-derived RNA biomarkers should decrease after surgical removal of the tumor and should be absent or have significantly lower levels in healthy females. Therefore, to identify BC-derived RNA biomarkers that could be exploited for the detection or monitoring of BC using liquid biopsies, we performed a deep sequencing analysis of the RNA cargo of plasma EVs collected at the time of diagnosis and after breast surgery from 32 patients with locally advanced BC and 30 cancer-free healthy controls (HCs) and searched for RNAs that had higher levels in BC EVs at the time of diagnosis compared to cancer-free controls and decreased after the surgery.

## 2. Materials and Methods

### 2.1. Study Population and Sample Collection

Patients diagnosed with stage II–III primary BC were enrolled in the study at Riga East University Hospital. The diagnosis was established by examining core needle biopsies. Patients who had a blood transfusion in the last six months or a history of another oncological disease were excluded from this study. The clinical and pathological features of the patients are shown in Table 1. The blood samples were collected at the time of diagnosis and 7 days after the surgical removal of the tumor. The processing and storage of blood samples have been described previously [27]. Plasma samples from 30 cancer-free age-matched females were obtained from the Latvian Genome Database.

The study protocol was approved by the Latvian Central Medical Ethics Committee (decision No. 1839). The clinical samples and information were collected after the patient’s informed written consent was obtained. 

### 2.2. EV Isolation, Quantification, and Quality Control

Isolation of EVs from plasma samples has been described before [27]. Briefly, EVs were isolated from 1 mL of plasma by size exclusion chromatography (SEC) using Sepharose CL2B (Cytiva, Marlborough, MA, USA). SEC fractions were measured with Zetasizer Nano ZS (Malvern, Worcestershire, UK) and fractions containing particles larger than 35 nm were collected and concentrated to 100 μL. All EV samples were quantified by nanoparticle tracking analysis with the NanoSight NS500 instrument (Malvern, UK). Selected samples were assessed by transmission electron microscopy (TEM) and Western blot (WB) analysis. For the TEM, a total of 10 µL of EV suspension in PBS was gently deposited onto a 300-mesh EM grid with a carbon-coated copper surface. Following a brief 5 min incubation, the samples underwent negative staining using a 1% uranyl formate solution for 60 s. Subsequent to air-drying, the samples were examined utilizing a JEM-1230 transmission electron microscope (JEOL, Peabody, MA, USA). 

For Western blot analysis, EVs were heat-treated at 95 °C for 5 min in the presence of reducing Laemmli buffer. An aliquot corresponding to 100 μL of plasma was loaded onto each lane of a 10% SDS-PAGE gel. Following separation at 150 V, the proteins were transferred onto 0.45 µm nitrocellulose membranes, which were subsequently blocked using 10% (*w*/*v*) fat-free milk for 1 h at room temperature. The membranes were then incubated overnight at +4 °C with primary antibodies against TSG101 (Abcam, Waltham, MA, USA, catalog #ab15011) at a dilution of 1:1000, Calnexin (Abcam, catalog #ab22595) at a dilution of 1:2000, or without antibody (for HRP-conjugated primary antibody) in 5% fat-free milk. Following appropriate washing steps, the membranes were incubated for 1 h at room temperature with either anti-rabbit IgG-HRP (Santa Cruz, Dallas, TX, USA, catalog #sc-3837) at a dilution of 1:2000, anti-mouse m-IgG BP-HRP (Santa Cruz, catalog #sc-516102) at a dilution of 1:2000, or HRP-conjugated antibody against CD63 (Novus Biologicals, Littleton, CO, USA, catalog #NBP2-34779H) at a dilution of 1:2000. Detection of immunoreactive bands was carried out using the Amersham™ ECL Select™ Western Blotting Detection Reagent kit manufactured by GE HealthCare Lifesciences (Chicago, IL, USA). The images were taken using a Nikon d610 digital single-lens reflex camera body (Nikon, Tokyo, Japan) combined with a Sigma 35 mm f/1.4 DG HSM Art lens (Sigma Corporation, Kanagawa, Japan).

### 2.3. Extraction of EV RNA and RNA Sequencing

Extraction of EV RNA and construction of RNA-seq libraries have been described before [27]. Briefly, the EV samples were treated with proteinase K (1 mg/mL) followed by RNAse A (10 ng/μL) and subjected to RNA extraction using miRNeasy Micro Kit (Qiagen, Germantown, MD, USA) according to the manufacturer’s protocol. RNA was eluted in 12 μL of RNAse-free water and the quality was checked using Agilent 2100 Bioanalyzer and RNA 6000 Pico Kit (Agilent Technologies, Santa Clara, CA, USA). A total of 2 μL of the EV RNA was used for the RNA-seq library construction with CleanTag^®^ Small RNA Library Prep Kit (Trilink Biotechnologies, San Diego, CA, USA). RNA-seq libraries were sequenced on Illumina NextSeq500 instrument using NextSeq 500/550 Mid Output Kit v2.5 (150 cycles) (Illumina, San Diego, CA, USA).

### 2.4. RNA-Seq Data Processing and Analysis

The raw data were acquired in FASTQ format and subjected to QC check and trimming of adapters using Cutadapt 4.2 [28]. The reads were mapped against Ensembl human genome (GRCh38) using Bowtie2 2.5.0 [29] and the multi-aligned reads were repositioned using ShortStack 3.8.5 [30]. The mapped reads were counted using Rsubread 2.14.2 package [31]. Annotations from miRbase, GtRNAdb, LNCipedia, lncRNAdb, piRBase, piRNABank, and piRNAdb databases were added to the GRCh38 annotation. For differentially expressed gene (DEG) analysis, the data were normalized to the count of reads mapped to the given RNA biotype and analyzed by quasi-likelihood F-tests using the edgeR 3.42.4 package. Multiple testing correction was performed by the Benjamini–Hochberg procedure and adjusted (adj.) *p*-value of ≤0.05 and absolute log_2_FC > 1 was considered to be significant. Differentially expressed mRNAs (adj. *p* < 0.05 and absolute Log_2_FC > 1) was subjected to GO term analysis using GOstats 2.66.0 and enrichment analyses using the rentrez 1.2.3 package, GO.db 3.17.0 package, and ShinyGO 0.75 package. Biomarker models were constructed using logistic regression analysis. The backward elimination approach was used to exclude less informative biomarkers. Leave-one-out cross-validation (LOOCV) was used to assess the robustness of the models. The RNA-seq datasets can be found in Array Express database: https://www.ebi.ac.uk/biostudies/arrayexpress/studies/E-MTAB-12014 (accessed on 24 October 2022).

### 2.5. Statistical Analysis

Statistical analysis was performed using RStudio 4.2 (RStudio Team, USA). For determining the statistical significance of differences in the read counts between independent groups of samples, Wilcoxon test was used and *p* value < 0.05 was considered significant. For multiple testing correction, the Benjamini–Hochberg procedure was used.

## 3. Results

### 3.1. Quality Control of EV Isolation

EVs were isolated from paired plasma samples collected at the time of diagnostic examinations (PreOp) and 7 days after mastectomy (PostOp) from 32 patients with locally advanced BC and 30 age-matched HCs. To assess the performance of the EV isolation protocol, EVs obtained from three randomly selected BC patients and three controls were characterized by TEM and WB analysis. TEM revealed that both BC and HC samples contain a mixture of vesicles in size range from 50 to 250 nm, and a large fraction of these vesicles have the cup-shape morphology that is characteristic to exosomes. Smaller particles (<40 nm in diameter) are also present that most likely represent lipoprotein particles; however, their fraction is relatively small, showing that the co-isolation of lipoproteins is not significant (Figure 1a,b). WB results showed that EVs were positive for typical EV markers CD63 and TSG101, and negative for calnexin, an endoplasmic reticulum protein, thus showing that the EV preparations do not contain significant contamination of ER membranes (Figure 1c). All EV preparations were routinely quantified by NTA that showed that the concentration of circulating EVs in BC patients at the time of diagnosis was significantly higher than in HCs—it ranged from 4.91 × 10^9^ to 3.68 × 10^11^ EVs/mL in BC patients, while, for HCs, the range was from 7.96 × 10^8^ to 8.46 × 10^10^ EVs per ml of plasma (*p* = 0.0061) (Figure 1d). However, we did not see any statistically significant difference between the EV concentrations in PreOp vs. PostOp samples (Figure 1e). The EV size ranged from 40 to 300 nm and no significant differences were found between the groups.

### 3.2. Composition of EV RNA Cargo in BC Patients and Cancer-Free Controls

We have shown before that more than 50% of the EV-associated RNA is attached to the surface of EVs [32]; therefore, in the current study, EVs were treated with proteinase K and RNAse A prior to the RNA extraction. Typical EV-RNA profiles obtained by Bioanalyzer are shown in Figure 2a,b. The profiles show a dominant RNA peak of 30–130 nt. No significant differences in the RNA profiles between BC patients before and after the surgery nor HCs were observed.

A total of 4.7 million raw reads were obtained per EV RNA-seq library and an average of 2.4 million reads passed the quality controls and read length filters. No statistically significant differences in the read count or length were observed between BC patients and HCs. An average of 1.3 million reads were mapped to the human genome version GRCh38. To assess the representation of various RNA biotypes in EVs, the reads mapped to overlapping features were prioritized in the following order: miRNAs > tRNAs > rRNA > mRNAs > pseudogenes > snRNAs > snoRNAs > piRNAs > lncRNAs > miscRNAs. The most common RNA biotypes found in EVs were lncRNA (28.3%) and mRNA (27.3%), followed by miRNA (18.6%) and piRNA (19.9%). A comparison of RNA biotype fractions revealed that the percentage of miRNAs, snRNAs, snoRNAs, and tRNA-derived fragments (tRFs) was higher, whereas the fraction of lncRNAs was lower in BC patients at the time of diagnosis than in HCs. In the PostOp samples, the distribution of miRNA and snRNA biotypes tended to become similar to the HCs (Figure 2c). 

### 3.3. Identification of mRNA Biomarker Candidates

A comparison of mRNA profiles between PreOp BC and HC EVs revealed 263 DEGs (adj. *p* < 0.05 and abs. Log_2_FC > 1). The majority of them—253—had higher levels in PreOp BC than in HC EVs (Figure 3a, Appendix A). GO term enrichment analysis showed that this gene set is strongly enriched in genes involved in translation and protein targeting to the endoplasmic reticulum, oxidative phosphorylation, and electron transport chain (Figure 3b). 

A comparison of PreOp EVs with PostOp EVs revealed 13 DEGs, 8 of which were decreased after the surgery, suggesting that they mainly originate from BC tissues (Figure 3c; Appendix A). To identify BC-derived RNAs with a diagnostic value, we searched for mRNAs that overlap in both DEG sets. This led to the identification of seven mRNAs that have higher levels in PreOp BC EVs as compared to HC EVs and a decrease in the PostOp EVs (Figure 3d; Table 2), thus representing biomarker candidates in liquid biopsies. All of them have a specificity of 1 as they were not detected in any HC, yet their sensitivity was below 0.19. Individually, they could distinguish PreOp BC from HC with AUCs of 0.55 to 0.59 (Figure 3e).

### 3.4. Identification of Biomarker Candidates in Non-Coding RNA Biotypes

We followed the same strategy as for mRNAs to identify cancer-derived biomarker candidates among the non-coding RNA biotypes. Differential expression analysis of miRNAs revealed 206 miRNAs with higher levels in PreOp BC EVs as compared to HC EVs and 43 miRNAs that decreased in the PostOp EVs as compared to PreOp EVs. Thirty-three of them overlapped, thus representing potential BC-derived biomarkers (Figure 4a; Appendix A). The top two miRNAs showing the highest diagnostic value were miR-224-5p and miR-200b-5p, which could distinguish BC from HCs with AUC of 0.72 and 0.70, respectively (Figure 4b).

lncRNA analysis revealed 382 lncRNAs that have higher levels in PreOp BC EVs than HC EVs, 25 lncRNAs whose levels decrease in the PostOp EVs, and 14 of them overlapped (Figure 4c; Appendix A). The top lncRNAs were lnc-IFT122-2 and lnc-GSR-2 and they could distinguish BC from HCs with AUC of 0.72 and 0.66, respectively (Figure 4d).

Likewise, 12 BC-derived biomarker candidates were identified in piRNA, 23 in snoRNA, 3 in snRNA, and 5 in tRF RNA biotypes (Figure 4e–h; Appendix A). Among them, SNORD3H and SNORD1C showed the highest diagnostic value with AUC of 0.75 and 0.73, respectively, followed by piR-27570 and piR-32949, both showing AUC of 0.67. The AUC for the best snRNA also was 0.67 (RNU4-11P) and 0.62 for the best tRF (Lys-TTT-4-1-tRF3T).

### 3.5. Construction of Biomarker Model

Next, we used logistic regression to construct a biomarker model for discriminating between PreOp BC and HC. The backward elimination approach was used to select a panel of most informative markers that resulted in the construction of an eight-RNA biomarker model. The model comprised three snoRNAs: SNORD3H, SNORD1C, and SNORA74D; three lncRNAs: lnc-IFT-122-2, lnc-C9orf50-4, and lnc-FAM122C-3; one miRNA: miR-224-5p; and one piRNA: piR-32949. On the training sample set, the model could distinguish BC from HC EVs with an AUC of 0.905 (95% CI = 0.872–0.939; *p* = 1.6 × 10^−8^), sensitivity of 0.82, and specificity of 1 (Figure 5a). The LOOCV was used to assess the robustness of the model and yielded AUC of 0.902 (95% CI = 0.872–0.931, *p* = 3.4 × 10^−9^), sensitivity of 0.88, and specificity of 0.90 (Figure 5b).

### 3.6. Association of RNA Biomarkers with Hormone Receptor Status

Finally, we assessed the association of the levels of EV RNAs at the time of diagnosis with various clinicopathological parameters and histological features of BC. Patients were dichotomized based on the clinical variables and the pretreatment levels of EV RNAs were compared between the groups of patients. Results showed that a number of RNA biomarkers from various biotypes were strongly associated with the ER and HER2 status. Next, the biomarker models for prediction of ER and HER2 status were constructed. A five-RNA model comprising Ala-CGC-2-1-tRF5D, SNORD116-4, snoRNA ACA64, miR-2467-5p, and piR-7447 could distinguish BC patients with ER+ from ER− tumors with an AUC of 0.924 (95% CI = 0.863–0.985, *p* = 9.2 × 10^−5^) on the training set and an AUC of 0.887 (95% CI = 0.809–0.966, *p* = 0.00038) in the LOOCV (Figure 5c,d). Similarly, a six-RNA model comprising lnc-MCL1-2, Val-AAC-5-1-tRF5D, RNU2-14P, piR-32971, SNORA36C, and LINC00567[−]/ING1-1[+] could distinguish between BC patients with HER2+ and HER2− tumors with an AUC of 0.976 (95% CI = 0.937–1.015, *p* = 4.8 × 10^−6^) on the training set and an AUC of 0.929 (95% CI = 0.8233–1.034, *p* = 3.2 × 10^−5^) in the LOOCV (Figure 5e,f).

## 4. Discussion

EVs are increasingly recognized as carriers of cancer-derived RNA biomarkers in liquid biopsies. In our previous study, we investigated the dynamics of EV levels and their RNA cargo during the treatment of BC patients and correlated the changes with the treatment response [27]. We showed that EV levels significantly increased during the chemotherapy and surgery and returned to the levels found in healthy individuals 6 months after surgical removal of cancer, suggesting that a substantial fraction of plasma EVs in BC patients are produced due to the disease-associated processes and treatment [27]. In this study, we examined the diagnostic significance of the EV RNA cargo in the same cohort of BC patients. 

An ideal cancer biomarker for liquid biopsies should be derived from cancer tissues, correlate with the status of the disease, and reflect the behavior of cancer. Typically, the tissue of origin for cancer biomarkers discovered in case–control studies remains unknown. In the current study, we interrogated the RNA cargo of plasma EVs collected from BC patients at the time of diagnosis and 7 days after the breast surgery and from cancer-free females. We reasoned that the levels of RNA biomarkers that are derived from cancer tissues should decrease after the surgical removal of the tumor; therefore, we searched for RNAs that have higher levels in BC patients than cancer-free controls and decrease after the surgery. We identified suitable candidates in all major RNA biotypes found in EVs. This approach led to the identification of biomarkers with very high specificity but moderate to poor sensitivity as they were detectable only in a fraction of BC patients. Hence, the diagnostic value for each individual biomarker is moderate. However, a biomarker model combining eight RNA biomarkers—three lncRNAs, three snoRNAs, one miRNA, and one piRNA could distinguish BC patients from cancer-free females with remarkably high AUC of 0.902. Although the LOOCV demonstrated a very high robustness of the model, it is still possible that the model is overfitted. The main limitation of our study is that we did not have an opportunity to test the model in an independent sample set. However, even if some of the markers fail in the validation set, there is an opportunity to replace them with other, equally well-performing markers and generate an alternative version of the model without compromising its performance. Furthermore, currently, RNAseq-based analyses are not suitable for routine clinical applications due to the high cost and poor reproducibility on different technological platforms; therefore, the next task would be establishing PCR-based assays for the biomarker candidates. We expect two main challenges with this task: at first, reproduction of RNAseq results by PCR is hampered by the fact that the global normalization approaches used for RNAseq data analyses are not applicable for the normalization of PCR data. Secondly, combining biomarkers representing different RNA biotypes in a single assay could be challenging due to different requirements for primer design and assay conditions. Another limitation of the current study is that we analyzed EVs from patients with locally advanced stage II to III BC that represent the largest part of newly diagnosed BC cases in Latvia, whereas no early-stage patients were recruited. Hence, we do not know what could be the diagnostic performance of the identified biomarkers for detecting stage I BC. 

Importantly, a number of RNA biomarkers correlated with ER and HER2 status of the tumor that would provide additional information about the properties of the tumor if an EV-based blood test was performed during the diagnostic examinations. This finding is consistent with a recent study showing that 17β-estradiol promotes EV secretion specifically from ER+ BC cells and regulates miRNA loading into EVs [33]. 

Although several cargo sorting mechanisms may lead to the enrichment or depletion of EVs with some specific RNAs, overall, the RNA content of EVs resembles that of their cell of origin [9]. Hence, the EV RNA cargo is expected to reflect the biological and pathological processes of their parental cells. We found that EVs collected from BC patients at the time of diagnosis were enriched in mRNAs involved in translation, protein targeting to the endoplasmic reticulum, mRNA catabolic processes, etc. They were also strongly enriched with mitochondrially encoded mRNAs such as NADH dehydrogenase subunits, ATP synthases, and cytochrome C oxidases that are not expected to be released into the cytoplasm of healthy cells, thus raising the question of how these mRNAs get packaged into EVs. A study by Su Chul Jang et al. (2019) showed that melanoma-tissue-derived EVs contained mitochondrial membrane proteins and these EVs could be detected in the plasma of patients with melanoma, ovarian, and breast cancer [34]. Another recent study described a novel mitochondria-derived EV subpopulation that is released by brain cells and showed that this pathway is altered by mitochondrial dysfunction [35]. Conceivably, the bulk EV population isolated from BC plasma contains mitochondria-derived EV subpopulation. However, the levels of mitochondrially encoded mRNAs did not decrease significantly in the PostOp EVs; hence, the tissue source of this EV subpopulation, the exact mechanism by which these EVs are generated, and their functional significance in cancer patients remain unknown.

miRNA is the most widely studied RNA biotype in EVs. A number of miRNAs identified as BC-derived biomarker candidates have been previously found in plasma EVs and/or have been implicated in the development or progression of BC. For example, miR-224-5p has been found in MSC-derived EVs and regulated proliferation, apoptosis, and autophagy in BC cells [36,37]. miR-200b-5p, a member of the miR-200 family, is involved in epithelial–mesenchymal transition and metastasis and its overexpression in BC tissues is associated with poor disease-free survival and overall survival [38]. High miR-200b-5p levels in plasma could distinguish metastatic from early BC and predicted shorter overall survival [39,40]. miR-218 was shown to be overexpressed by metastatic BC cells and its plasma level was associated with BC bone metastasis [41]. Interestingly, miR-485-5p and miR-133 have been shown to function as tumor suppressors in several types of cancer, including BC [42,43]. Thus, increased levels of these miRNAs in EVs from BC patients support the idea that cancer cells actively secrete tumor suppressor miRNAs via sorting them into EVs in order to decrease their intracellular level [33].

Other non-coding RNA biotypes such as snoRNAs, snRNAs, piRNAs, and tRFs are just emerging as potential cancer biomarkers. Our study shows that the fraction of snoRNAs, snRNAs, and tRFs in the EV RNA pool is increased in BC patients and revealed a number of BC-derived biomarker candidates in each of these biotypes. snoRNAs are 60–300 nt long non-coding RNAs that are classified into three major groups depending on their length, secondary structure, and biological functions: SNORDs, SNORAs, and SCARNAs. The canonical function of snoRNAs is guiding 2′-O-methylation and pseudouridylation of ribosomal RNAs that is required for the maturation and stabilization of rRNAs; however, it is now becoming clear that they are involved in functions distinct from the nucleolus, such as alternative splicing, mRNA stability, and regulation of gene expression [44]. Emerging evidence suggests that snoRNAs contribute to tumorigenesis and metastasis when mutated or aberrantly expressed in cancer [45,46]. In our study, SNORD3H and SNORD1C showed the highest diagnostic value among all biomarker candidates and were the most significant components of the diagnostic biomarker model, whereas SNORD116-4 and SNORA36C levels were associated with hormone receptor status. To the best of our knowledge, this is the first study suggesting that EV-enclosed snoRNAs may be exploited for the detection of BC, though the diagnostic value of circulating snoRNAs in pancreatic [47], lung [48], and renal cancer [49] has been reported before. 

snRNAs are components of spliceosomes and abnormal expression of snRNAs in cancer drives gene-specific alterations in the splicing pattern and contributes to the establishment of global splicing programs [50]. Here, we found three snRNAs (RNU1-40P, RNU1-16P, and RNU-11P) with potential diagnostic significance; however, the role of these specific RNAs in BC has not been reported so far. 

tRFs are tRNA-derived fragments generated by enzymatic cleavage of mature tRNAs or tRNA precursors at specific positions. These fragments are functional and are associated with diverse human diseases, including cancer [51]. They have been shown to promote ribosome biogenesis, participate in the regulation of transcription, and inhibit protein translation via binding to RNA-binding proteins, as well as regulating reverse transcription, nascent RNA silencing, and mRNA cleavage via complementary base pairing and guiding Argonaute proteins [51,52,53]. Cell-free tRFs and tRNA halves (another type of tRNA-derived fragment) have been detected in the bloodstream, both associated with EVs and in a vesicle-free form [54]. We found that the fraction of tRNAs and various tRNA-derived fragments is significantly increased in the plasma EVs from BC patients as compared to healthy controls and five specific tRFs showed a potential diagnostic value. One of them, Lys-TTT-4-1-tRF3T, has been previously found to be increased in serum EVs from early-stage BC patients as compared to healthy controls [55]. 

## 5. Conclusions

This study showed that the proportions of various RNA biotypes in plasma EVs from BC patients are altered and BC-derived RNA biomarker candidates were identified among mRNAs, miRNAs, lncRNAs, snoRNAs, piRNAs, snRNAs, and tRFs, thus suggesting that they represent a rich yet unexplored source of cancer biomarkers. Considered individually, these biomarkers had very high specificity but a moderate sensitivity, whereas a biomarker model composed of eight RNA biomarkers could detect BC with high accuracy and potentially can be exploited for the detection and monitoring of BC using liquid biopsies. Additional sets of biomarkers were established that correlate with the ER and HER2 status. The next steps toward translation of these results into clinically applicable tools will be to develop PCR-based assays for the candidate biomarkers and test them in independent sample sets.

## Figures and Tables

**Figure 1 cancers-15-04329-f001:**
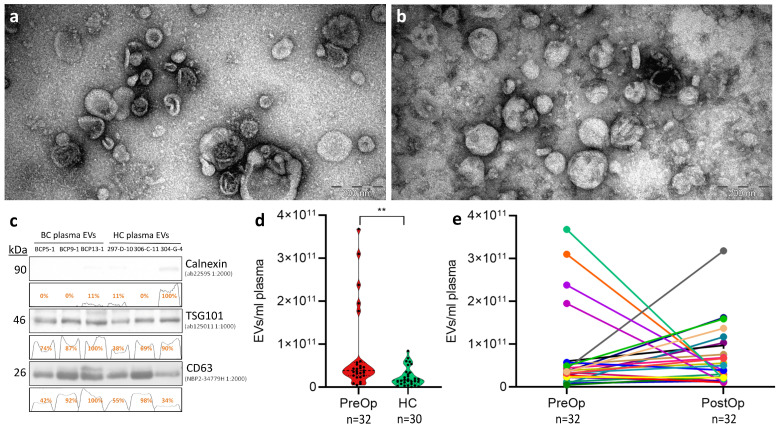
Quality control and quantification of extracellular vesicles. Transmission electron microscopy images of EVs isolated from plasma of BC patient (**a**) and cancer-free healthy control (**b**). Scale bar is 200 nm; (**c**) Western blot analysis of EV markers (TSG101 and CD63) and an endoplasmic reticulum protein Calnexin (negative control) in EVs isolated from plasma of BC patients and cancer-free healthy controls; The uncropped blots are shown in Appendix A. (**d**) violin plots showing EV concentrations per ml of plasma in BC patients at the time of diagnosis and HCs. (**e**) Paired dot plot showing the comparison of plasma EV levels in BC patients at the time of diagnosis and after surgical removal of the tumor. Statistical significance was determined with Wilcoxon test and *p* < 0.05 was considered significant. PreOp, diagnostic time point; HC, healthy controls; PostOp, 7 days after mastectomy; **, *p* value < 0.05.

**Figure 2 cancers-15-04329-f002:**
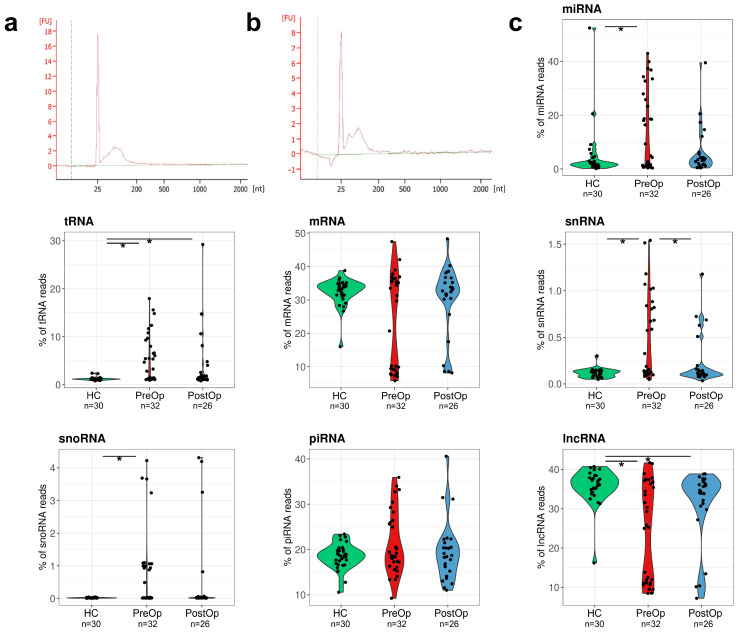
Composition of EV RNA cargo. (**a**) The RNA size distribution in plasma EVs from a BC patient and (**b**) a healthy control visualized with Bioanalyzer RNA Pico Chip; (**c**) violin plots showing the percentage of each RNA biotype in the plasma EV samples. Statistical significance was determined with Wilcoxon test and *p* value < 0.05 was considered significant. PreOp, diagnostic time point; HC, healthy controls; PostOp, seven days after breast surgery; *, *p* value < 0.05.

**Figure 3 cancers-15-04329-f003:**
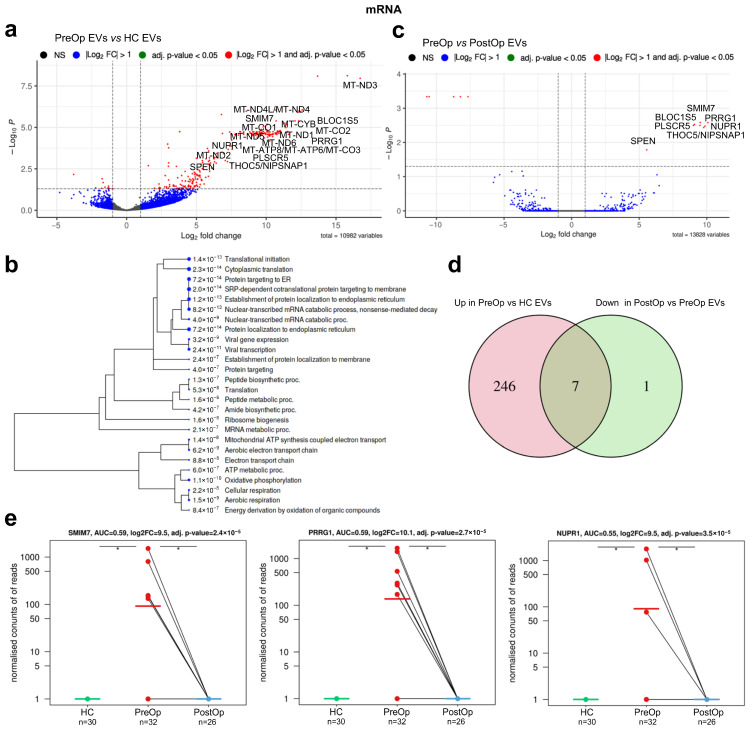
Identification of mRNA biomarker candidates. (**a**) Volcano plot showing differentially expressed mRNAs between EVs isolated from BC plasma at the time of diagnosis and healthy controls; (**b**) GO term enrichment analysis in EVs isolated from BC plasma. A hierarchical clustering tree summarizing the correlation among significant pathways represented by DEGs in BC EVs at the time of diagnosis vs. healthy controls. Pathways with many shared genes are clustered together. Bigger dots indicate more significant *p* values; (**c**) volcano plot showing differentially expressed mRNAs between EVs isolated from BC plasma at the time of diagnosis and after breast surgery; (**d**) Venn diagram showing the overlap of DEGs that are upregulated in diagnostic samples and decreased after the surgery; (**e**) dot plots showing the EV levels (normalized read counts) of selected mRNA biomarker candidates in HCs and BC patients at the time of diagnosis and after surgery. HC, healthy control; PreOp, diagnostic time point; PostOp, seven days after breast surgery; *, adj. *p* value < 0.05.

**Figure 4 cancers-15-04329-f004:**
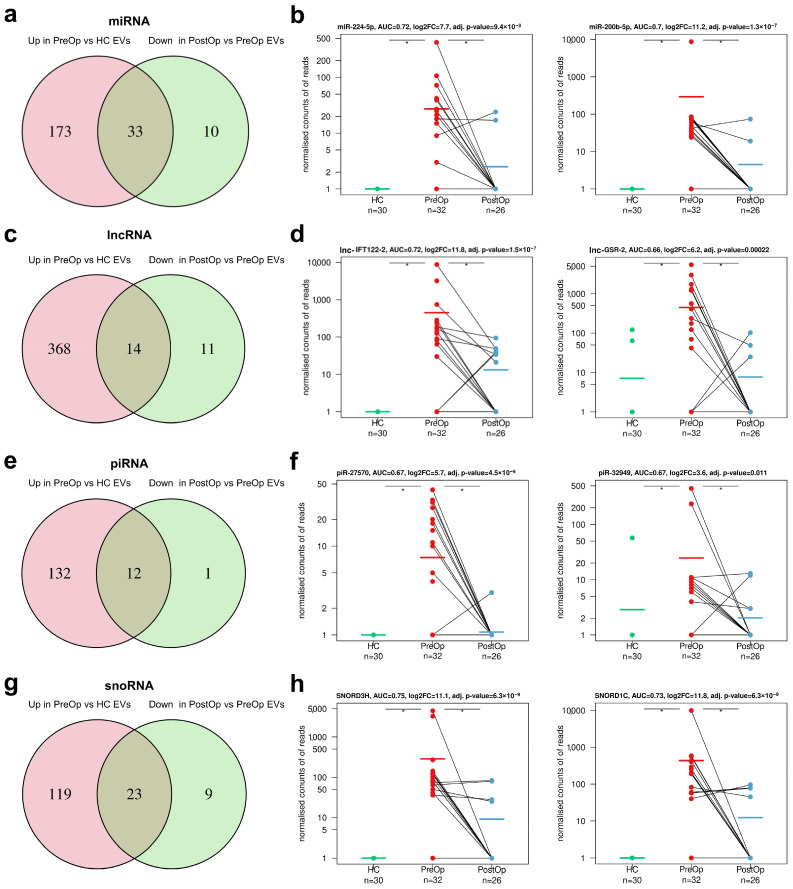
Identification of RNA biomarker candidates in non-coding RNA biotypes. Venn diagrams show the number of DEGs with higher levels in PreOp BC EVs as compared to HC EVs and decreased levels in the PostOp EVs as compared to PreOp EVs. Dot plots showing the EV levels (normalized read counts) of selected RNA biomarker candidates in HCs and BC patients at the time of diagnosis and after surgery. (**a**,**b**) miRNAs; (**c**,**d**) lncRNAs; (**e**,**f**) piRNA; (**g**,**h**) snoRNA. HC, healthy control; PreOp, diagnostic time point; PostOp, seven days after breast surgery; *, adj. *p* value < 0.05.

**Figure 5 cancers-15-04329-f005:**
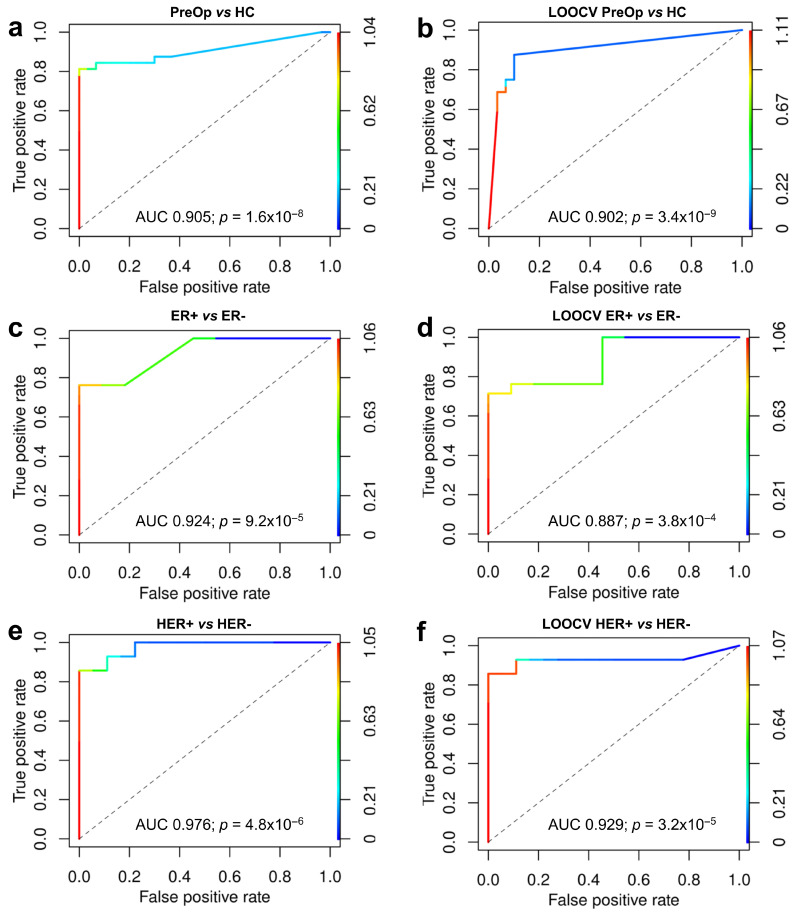
Biomarker models. (**a**) ROC curve for 8-RNA biomarker model that distinguishes between BC patients and healthy controls; (**b**) leave-one-out cross-validation (LOOCV) of the BC vs. HC biomarker model; (**c**) ROC curve for 5-RNA biomarker model that distinguishes between BC patients with ER+ vs. ER− tumor; (**d**) LOOCV of the ER+ vs. ER− biomarker model; (**e**) ROC curve for 6-RNA biomarker model that distinguishes between BC patients with HER2+ vs. HER2− tumors; (**f**) LOOCV of the HER2+ vs. HER2 biomarker model. ER, estrogen receptor; PreOp, diagnostic time point; HC, healthy control.

**Table 1 cancers-15-04329-t001:** Clinical characteristics of the study population.

Characteristic	BC Patients
Sample size (n)	32
Age mean, years	50.9
Age range, years	34–77
**Tumor grade**	
Grade 2	21
Grade 3	11
**TNM stage**	
T1 N1-3 M0	2
T2 N1-3 M0	12
T3 N1-3 M0	16
T4 N1-3 M0	2
**Estrogen receptor**	
Positive	20
Negative	12
**Progesterone receptor**	
Positive	17
Negative	15
**HER2 overexpression**	
0	5
1	13
2	4
3	10
**TNBC**	
Yes	8
No	24
**E-cadherin**	
Positive	22
Negative	10
**Proliferation index (Ki-67)**	
≤14%	5
>14%	27

**Table 2 cancers-15-04329-t002:** Potential mRNA biomarkers for the detection of BC.

Gene	Full Name	Log_2_FC PreOp vs. HC	Adj. *p* Value	Log_2_FC PreOp vs. PostOp	Adj. *p* Value	Function *
*SMIM7*	small integral membrane protein 7	9.508	2.35 × 10^−5^	9.508	0.0026	Predicted to be integral component of membrane
*PRRG1*	proline rich and Gla domain 1	10.086	2.66 × 10^−5^	10.086	0.0026	Enables calcium ion and protein binding
*BLOC1S5-TXNDC5*	biogenesis of lysosomal organelles complex 1 subunit 5 BLOC1S5-TXNDC5 readthrough (NMD candidate)	9.090	2.71 × 10^−5^	9.090	0.0030	BLOC1S5 enables protein binding, involved in the biogenesis of organelles. BLOC1S5-TXNDC5 is a naturally occurring read-through transcription between the neighboring *MUTED* and *TXNDC5* genes on chromosome 6. A candidate for nonsense-mediated mRNA decay (NMD) and is unlikely to produce a protein product
*PLSCR5*	phospholipid scramblase family member 5	9.007	3.11 × 10^−5^	9.007	0.0032	Predicted to enable phospholipid scramblase activity and be involved in plasma membrane phospholipid scrambling
*NUPR1*	nuclear protein 1, transcriptional regulator	9.494	3.50 × 10^−5^	9.494	0.0032	Enables DNA binding activity and transcription coactivator activity. Involved in regulation of cellular catabolic process; regulation of generation of precursor metabolites and energy; and regulation of programmed cell death. Acts upstream of or within negative regulation of cell cycle.
*SPEN*	spen family transcriptional repressor	5.558	0.0002	5.558	0.0165	Hormone inducible transcriptional repressor
*THOC5/NIPSNAP1* (reads overlap both genes)	THO complex 5/nipsnap homolog 1	9.858	4.23 × 10^−5^	9.858	0.0033	THOC5 is predicted to enable mRNA binding activity. Involved in monocyte differentiation, negative regulation of DNA damage checkpoint, and viral mRNA export from host cell nucleus. NIPSNAP1 may be involved in vesicular transport. A similar protein in mice inhibits the calcium channel TRPV6, and is also localized to the inner mitochondrial membrane where it may play a role in mitochondrial DNA maintenance.

* Information about gene function from NCBI Gene.

## Data Availability

The RNAseq datasets are available at ArrayExpress, accession number/identifier: E-MTAB-12014. Permanent link: https://www.ebi.ac.uk/arrayexpress/experiments/E-MTAB-12014 (accessed on 24 October 2022).

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
