# Peer review of "Extracellular Vesicles—A Source of RNA Biomarkers for the Detection of Breast Cancer in Liquid Biopsies"

_cancers, 2023, doi:10.3390/cancers15174329_

Round 1
Reviewer 1 Report
This study is interesting with clinical significance. Extracellular Vesicle is an important factor in intercellular communication and play an important in tumor metastasis . The authors made a comprehensive study of Extracellular Vesicles. The followings are some comments to the authors.
Comments:
1.Please keep the abbreviations consistent . For example, in line 26 “cancer-free females (HCs)”, in line 159 cancer-free healthy females (HC).
2.Whether the expression of genes in Figure 3e and Figure 4b,d,f,h are different in different tumor grades or stage. I suggest further data analysis should be done. It is helpful to identify the genes associated with tumor development.
3. I suggest the effects of genes in Figure 3e and Figure 4b,d,f,h on the proliferation of breast cancer cell lines should be added. I think this helps demonstrate the role of these genes.
4.The discussion and conclusion can be improved. These kinds of studies have limitations. Hence, the author should have stated the potential limitations and suggested what could be done the next step in this area of research and application of EVs as a source in liquid biopsies.
Author Response
Dear Reviewer,
Thank you very much for your time, effort and valuable comments!
- We corrected the abbreviations to keep them consistent.
- Thank you for the suggestion. Indeed, we compared the levels of the identified biomarkers in plasma EVs from BC patients with grade 2 vs grade 3 tumors, and in T1-2 N1-3M0 vs T3-4 N1-3 M0 stages. Although some biomarkers tended to have different levels in groups of patients divided by the TNM stage, none of them remained significant after multiple testing correction.
- It is certainly of great interest to study the functional significance of the identified RNA biomarkers, however this is beyond the scope of the current study.
- We have added comments on the limitations of the current study and next steps toward clinical translation of the identified biomarkers (lines 352-368; 452-454).
Reviewer 2 Report
Extracellular vesicles are well-known mediators in the development of cancers and are promising in early detection. Plenty of studies have demonstrated their tremendous value in cancer prevention. However, the clinical applications are limited due to the lack of specific markers. The current research discussed an approach to detect breast cancer via bioliquid species, which might interest the readers. The authors should address the following questions before they can be published.
Major revisions:
1. many articles have discussed the roles of extracellular vesicles in diagnosing, preventing, and subtyping breast cancers. The authors should discuss this in the introduction section and emphasis the priorities of this study.
2. Mammography is the most common and efficient screening method for breast cancer, including early stage. In this paper, the samples were collected from patients with stage II-III BC and the EVs were further analyzed. What is the RNA expression pattern in EV samples from early-stage BC? I think this would be more interesting.
3. To confirm whether the DEGs in EVs were from breast cancer cells, the author should detect the expressions of those DEGs in the breast cancer tissues.
4. What is the percentage of cancer-derived EVs in the plasma samples? Is there a plan to address this question in the future?
Minor revisions:
1. Please add the protein ladder in the unprocessed WB images.
2. Please provide detailed information for TEM and WB procedures.
3. Please add the sample size to Figure 1d and 1E. Does the number of black dots in Figure 1d correspond to the sample number?
4. Please add the sample size to the subpanels of Figure 2.
5. Please add the statistical method in the Methods section.
Author Response
Dear Reviewer,
Thank you very much for your time, effort and valuable comments!
- We revised the Introduction and tried to emphasize the novelty of this study (lines 83-102).
- We agree that the lack of early-stage BC patients is a clear limitation of this study and we included a comment on this in the Discussion (lines 364-368).
- We partially agree to the point. We believe that the fact that the level of a given RNA biomarker decrease after surgical removal of tumour suggests that its presence in PreOp samples is related to the presence of tumour. Of course, validation of its expression in tumour tissues would further support its cancer-associated origin. In fact, we have performed full transcriptome analysis of cancer and adjacent normal breast tissues from 10 BC patients and could detect the expression of the majority of candidate biomarkers in the tumour tissues. However, for some of the patients with the highest levels of candidate biomarkers in plasma EVs, the tissue samples were not available, hampering the correlation of the biomarker levels in EVs and tissues, therefore we decided not to include these data in the current manuscript.
- The percentage of the cancer-derived EVs in plasma is a very important question, however, the currently available EV isolation techniques does not allow reliable separation of cancer-derived EVs from normal cell-derived EVs. For this, deeper knowledge about the properties of cancer-derived EVs (different surface molecules, membrane structure or physical properties ect. ) and novel technologies for their separation are needed.
Minor points:
- We added the protein ladder to the unprocessed WB images.
- We provided detailed descriptions of TEM and WB procedures in the Methods (lines 124-145).
- We provided sample size in Figure 1d and 1e.
- We provided sample size in Figure 2.
- We added Statistical analysis in the Methods section.